# cnm-positive *Streptococcus mutans* is associated with galactose-deficient IgA in patients with IgA nephropathy

Taro Misaki[1,2]*, Shuhei Naka[3], Hitoshi Suzuki[4], Mingfeng Lee[4], Ryosuke Aoki[4], Yasuyuki Nagasawa[5], Daiki Matsuoka[3], Seigo Ito[6], Ryota Nomura[7,8], Michiyo Matsumoto-Nakano[3], Yusuke Suzuki[4], Kazuhiko Nakano[8]

1 Division of Nephrology, Seirei Hamamatsu General Hospital, Hamamatsu, Shizuoka, Japan, 2 Department of Nursing, Faculty of Nursing, Seirei Christopher University, Hamamatsu, Shizuoka, Japan, 3 Department of Pediatric Dentistry, Okayama University Graduate School of Medicine, Dentistry and Pharmaceutical Sciences, Okayama, Okayama, Japan, 4 Department of Nephrology, Juntendo University Faculty of Medicine, Hongo, Bunkyo-ku, Tokyo, Japan, 5 Department of General Internal Medicine, Department of Internal Medicine, Hyogo College of Medicine, Nishinomiya, Hyogo, Japan, 6 Department of Internal Medicine, Japan Self-Defense Iruma Hospital, Iruma, Saitama, Japan, 7 Department of Pediatric Dentistry, Graduate School of Biomedical and Health Sciences, Hiroshima University, Hiroshima, Japan, 8 Division of Oral Infection and Disease Control, Department of Pediatric Dentistry, Osaka University Graduate School of Dentistry, Suita, Osaka, Japan

* misakitar@gmail.com

**Data Availability Statement:** All relevant data are within the manuscript and its Supporting Information files.

## Abstract

The presence of *Streptococcus mutans* expressing Cnm protein encoded by *cnm* (*cnm*-positive *S. mutans*) in the oral cavity is associated with immunoglobulin A (IgA) nephropathy (IgAN). However, the precise mechanism by which *cnm*-positive *S. mutans* is involved in the pathogenesis of IgAN remains unclear. The present study evaluated glomerular galactose-deficient IgA1 (Gd-IgA1) to clarify the association between the presence of *cnm*-positive *S. mutans* and glomerular Gd-IgA1 in patients with IgAN. The presence of *S. mutans* and *cnm*-positive *S. mutans* was evaluated by polymerase chain reaction in saliva specimens from 74 patients with IgAN or IgA vasculitis. Immunofluorescent staining of IgA and Gd-IgA1 using KM55 antibody in clinical glomerular tissues was then performed. There was no significant association between the glomerular staining intensity of IgA and the positive rate of *S. mutans*. However, there was a significant association between the glomerular staining intensity of IgA and the positive rate of *cnm*-positive *S. mutans* ($P < 0.05$). There was also a significant association between the glomerular staining intensity of Gd-IgA1 (KM55) and the positive rate of *cnm*-positive *S. mutans* ($P < 0.05$). The glomerular staining intensity of Gd-IgA1 (KM55) was not associated with the positive rate of *S. mutans*. These results suggest that *cnm*-positive *S. mutans* in the oral cavity is associated with the pathogenesis of Gd-IgA1 in patients with IgAN.

**Funding:** This work was supported by Japan Society for the Promotion of Science (grant numbers 19K10098, Dr. Taro Misaki, 21H03149, Dr Kazuhiko Nakano and 21K08242, Dr Yasuyuki Nagasawa). The sponsors or funders did not play any role in the study design, data collection and analysis, decision to publish, or preparation of the manuscript.

**Competing interests:** The authors have declared that no competing interests exist.

## Introduction

Immunoglobulin A (IgA) nephropathy (IgAN) is the most prevalent type of primary glomerulonephritis worldwide [1, 2]. Approximately 30% of renal biopsy cases involve IgAN [3] and 30% to 40% of patients with IgAN progress to end-stage kidney disease within 20 years [1, 2]. In renal biopsy specimens, IgA1, but not IgA2, is predominantly deposited in the mesangial and peripheral capillary regions [4].

Although the precise mechanism is still unknown [5], several studies have suggested that galactose-deficient IgA1 (Gd-IgA1) is a key effector molecule in the pathogenesis of IgAN [6–9]. A novel lectin-independent enzyme-linked immunosorbent assay (ELISA) using an anti-Gd-IgA1 monoclonal antibody (KM55) was recently developed [10]. Glomerular Gd-IgA1 deposition has been shown by immunofluorescence with KM55 antibody, which provides new insights into the possibility that Gd-IgA1 functions as a key effector molecule of IgAN [10, 11]. Gd-IgA1 has been specifically detected in IgAN and IgA vasculitis, but not in other renal diseases [11].

Patients with IgAN sometimes present with macroscopic hematuria if they develop an upper respiratory tract infection, such as tonsillitis [12]. Several bacterial species have been reported to be potential contributors to the pathogenesis of IgAN [13–17], including periodontitis-related [18, 19] and dental caries-related [20–27] bacteria. *Streptococcus mutans*, which is a Gram-positive oral streptococcal species, is a major pathogen causative of dental caries [28]. *S. mutans* with the *cnm* gene (*cnm*-positive *S. mutans*), which encodes Cnm (collagen-binding cell surface protein) [29], can bind to the extracellular matrix [30]. Therefore, Cnm protein is considered as a virulence factor, such as in infective endocarditis [31], inflammatory bowel disease [32], cerebral hemorrhage [33–35], and non-alcoholic steatohepatitis [36, 37]. Recent clinical studies have suggested that *cnm*-positive *S. mutans* is associated with IgAN [20–22, 25, 26]. The results of recent studies in animal models also support this conclusion [23, 24]. Overall, the findings to date suggest that *cnm*-positive *S. mutans* is an important pathogen in IgAN.

Whether *cnm*-positive *S. mutans* is related to the production of Gd-IgA1 when IgAN is induced is unknown. If *cnm*-positive *S. mutans* proves to be associated with Gd-IgA1, this is a major discovery and could be a breakthrough in determining the pathogenic mechanism of IgAN. Although an association between bacterial infection and IgAN is assumed [13–17], there is no evidence that Gd-IgA1 production is involved in the mechanism of IgAN induced by infection.

In the present study, we analyzed Gd-IgA1 using KM55 antibody to clarify the association between the presence of *cnm*-positive *S. mutans* in the oral cavity and glomerular Gd-IgA1.

## Materials and methods

### Patients and clinical characteristics

Seventy-eight patients who underwent a renal biopsy at Seirei Hamamatsu General Hospital, Hamamatsu, Japan in 2017–2021 were initially included. All patients were over 18 years of age. These patients were diagnosed with IgAN (n = 68) or IgA vasculitis (Henoch–Schonlein purpura nephritis) (n = 10) by renal biopsies. Their diagnoses were made on the basis of light microscopy and immunohistochemical findings. Four of the 78 patients were excluded because there were no glomeruli in prepared sections for Gd-IgA1 (KM55) staining. Therefore, analysis was performed on 74 patients. Clinical data (age, sex, height, body weight, body mass index, blood pressure, serum creatinine, estimated glomerular filtration rate [eGFR], serum C3, serum IgA, urinary protein [g/g creatinine], and urinary sediment of red blood cells

[RBCs] [>100/high-power field]) were obtained at the time of renal biopsy after patients had provided informed consent to participate in this study.

## Analysis of *cnm*-positive *S. mutans*

Saliva specimens obtained from the patients and frozen at –20˚C were used to test for the presence of *S. mutans* DNA and whether it was *cnm*-positive or *cnm*-negative by polymerase chain reaction, as described previously [21]. *Campylobacter rectus* and *Porphyromonas gingivalis* DNA was also evaluated, as described elsewhere [38].

## Histological studies

Renal biopsy specimens were collected via percutaneous needle biopsy. Paraffin-embedded 3-μm thick sections of renal specimens were stained with periodic acid Schiff, silver methenamine, and Masson trichrome. For the immunofluorescence analysis, frozen sections were subjected to fluorescence by fluorescein-conjugated goat IgG fraction to human IgG (F110FC, American Qualex, San Clemente, CA, USA), fluorescein-conjugated goat IgG fraction to human IgA (55077, MP Biomedicals, Solon, OH, USA), fluorescein-conjugated goat IgG fraction to human IgM (55153, MP Biomedicals), fluorescein-conjugated goat IgG fraction to human C3 (55167, MP Biomedicals), fluorescein-conjugated rabbit anti-human C1q (F0254, DAKO Japan Inc., Kyoto, Japan), and fluorescein-conjugated goat IgG fraction to human fibrinogen (55169, MP Biomedicals). IgAN or IgA vasculitis was diagnosed on the basis of mesangial cell proliferation in light microscopic findings, mesangial IgA deposition in immunofluorescence findings, and mesangial electron dense deposits in electron microscopic findings. Histological findings were evaluated according to the Oxford classification [39–41]. The Oxford classification of IgAN includes the following five histological variables: mesangial hypercellularity (M0/M1 lesion), segmental glomerulosclerosis (S0/S1 lesion), endocapillary hypercellularity (E0/E1 lesion), tubular atrophy/interstitial fibrosis (T0/T1/T2 lesion), and crescents (C0/C1/C2 lesion) [41]. We compared the renal histology in the *cnm*-positive *S. mutans* group with that in the *cmn*-negative *S. mutans* group using the Oxford classification in a blind test.

## Immunofluorescent staining of Gd-IgA1

Immunofluorescent staining of Gd-IgA1 and IgA in glomerular tissues obtained by renal biopsy was evaluated as described previously [11]. Paraffin-embedded 3 μm thick sections of the renal specimens were used for staining. Anti-human Gd-IgA1 antibody (KM55, Immuno-Biological Laboratories Co., Ltd, Gunma, Japan; 100 μg/ml) was used to evaluate Gd-IgA1, Alexa Fluor 555-conjugated goat anti-rat IgG antibody (1:1000; Life Technologies, Carlsbad, CA, USA) for the secondary antibody, and fluorescein-conjugated polyclonal rabbit anti-human IgA antibody (DAKO Japan; 100 μg/ml) for IgA [11]. The intensity of glomerular Gd-IgA1 and IgA was scored semiquantitatively (0–3 intensity) in a blind test [11].

## Measurement of Gd-IgA1

Plasma was collected at the time of the renal biopsy. Plasma Gd-IgA1 concentrations were evaluated using the KM55 ELISA kit (Immuno-Biological Laboratories Co., Ltd). Briefly, the ELISA plates were incubated with plasma specimens (1:200 dilution in enzyme immunoassay buffer) and standard specimens for 1 hour at room temperature, washed 4 times with wash buffer, incubated with prepared-labeled antibodies, and then treated with 50 ml of tetramethylbenzidine solution for 30 minutes in the dark. Absorbance was evaluated at 450/650

nm by the Versamax Microplate Reader (Molecular Devices, Tokyo, Japan). The Gd-IgA1 concentrations was evaluated according to the standard curve.

### Statistical analysis

All the results are expressed as the mean ± standard deviation (SD). When there was a significant difference, a further statistical analysis was conducted using Fisher's PLSD test or Fisher's exact test between 2 groups. The Cochran–Armitage trend test was used for trend analysis. A simple regression analysis was used for correlation analysis. In these analyses, $P < 0.05$ was considered statistically significant. The statistical analyses were conducted using Statview software (SAS Institute Inc., Cary, NC, USA) and GraphPad Prism 8 software (San Diego, CA, USA).

## Results

### Glomerular staining intensity of IgA and Gd-IgA1

The glomerular staining intensity of IgA and Gd-IgA1 was defined by the intensity of 0 to +3 as shown in Fig 1. IgA and Gd-IgA1 were mainly positive in the mesangial region, and IgA and Gd-IgA1 had the same distribution. There was a significant association between the glomerular staining intensity of IgA and Gd-IgA1 using regression analysis (R = 0.787, $P < 0.001$).

Plasma Gd-IgA1 concentrations were measured in 33 of 74 patients. There was no significant association between the glomerular staining intensity of Gd-IgA1 and plasma Gd-IgA1 concentrations (n = 33) (S1 Fig).

### Association between glomerular staining intensity of IgA and Gd-IgA1, and the rate of *cnm*-positive *S. mutans* in the oral cavity

There was no significant association between the glomerular staining intensity of IgA and the positive rate of *S. mutans* (Fig 2A). However, there was a significant association between the glomerular staining intensity of IgA and the positive rate of *cnm*-positive *S. mutans* (Fig 2B). There was also no significant association between the glomerular staining intensity of Gd-IgA1 and positive rate of *S. mutans* (Fig 2C). However, there was a significant association between the glomerular staining intensity of Gd-IgA1 and the positive rate of *cnm*-positive *S. mutans* (Fig 2D).

There was no significant association between the glomerular staining intensity of IgA and the positive rate of *C. rectus* (S2A Fig) or between the glomerular staining intensity of Gd-IgA1 and positive rate of *C. rectus* (S2B Fig). Furthermore, there was no significant association between the glomerular staining intensity of IgA and the positive rate of *P. gingivalis* (S2C Fig) or between the glomerular staining intensity of Gd-IgA1 and the positive rate of *P. gingivalis* (S2D Fig).

### Comparison of the *cnm*-positive *S. mutans* and *cnm*-negative *S. mutans* groups

The percentage of the staining intensity of IgA: 3+ was significantly higher in the *cnm*-positive *S. mutans* group than in the *cnm*-negative *S. mutans* group (Table 1). The percentage of the staining intensity of Gd-IgA1: 3+ was also significantly higher in the *cnm*-positive *S. mutans* group than in the *cnm*-negative *S. mutans* group (Table 1). No significant differences were found in age, sex, height, body weight, body mass index, systolic blood pressure, diastolic blood pressure, serum creatinine concentrations, eGFR, serum C3 concentrations, serum IgA

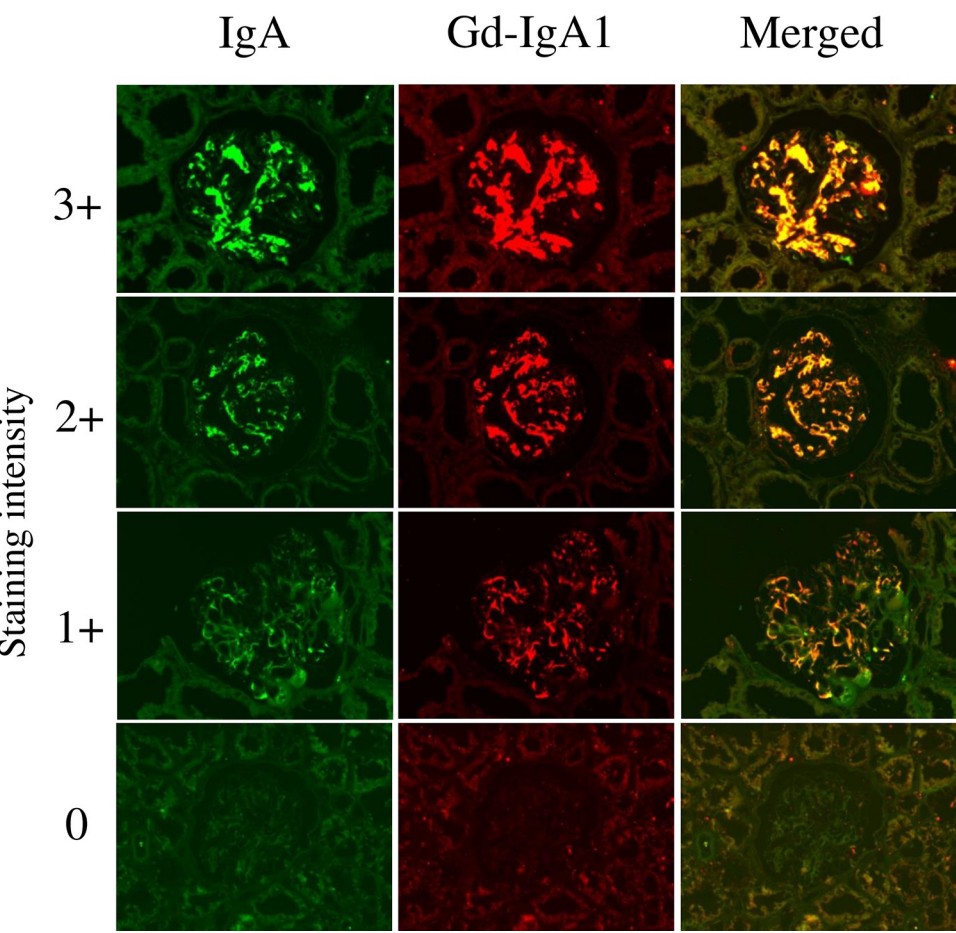

**Fig 1. Glomerular staining intensity of IgA and galactose-deficient IgA1.** Double staining with anti-IgA polyclonal antibody and Gd-IgA1 (KM55) monoclonal antibody was performed on biopsy specimens. Representative images of glomerular IgA and Gd-IgA1 (KM55) in patients with 0, 1+, 2+, and 3+ staining intensity. Seventy-four patients were divided into four groups according to the staining intensity of IgA: group 0 (n = 4), group 1+ (n = 46), group 2+ (n = 12), and group 3+ (n = 12). These patients were also divided into four groups according to the staining intensity of Gd-IgA1: group 0 (n = 4), group 1+ (n = 35), group 2+ (n = 23), and group 3+ (n = 12). Fluorescein-conjugated polyclonal rabbit anti-human IgA antibody (DAKO Japan Inc., Kyoto, Japan), anti-human Gd-IgA1 antibody (KM55) (Immuno-Biological Laboratories Co., Ltd, Gunma, Japan), and Alexa Fluor 555-conjugated goat anti-rat IgG antibody (Life Technologies) was used for immunofluorescence staining. Original magnification: ×200. Gd-IgA1, galactose-deficient IgA1.

concentrations, or urinary protein concentrations between the groups (Table 1). The percentage of urinary sediment ≥100 RBCs/high-power field was higher in the *cnm*-positive *S. mutans* group than in the *cnm*-negative *S. mutans* group, but this was not significant (Table 1). No significant differences were found in the mesangial hypercellularity score, endocapillary hypercellularity score, segmental glomerulosclerosis score, tubular atrophy/interstitial fibrosis score, or the crescent score of the Oxford classification between the groups (Table 2).

## Discussion

As far as we know, this is the first study to demonstrate an association between *cnm*-positive *S. mutans* in the oral cavity and glomerular Gd-IgA1. We discovered that there was a significant association between the positive rate of *cnm*-positive *S. mutans* in the oral cavity and glomerular staining intensity of IgA or Gd-IgA1. Although various factors are considered to be

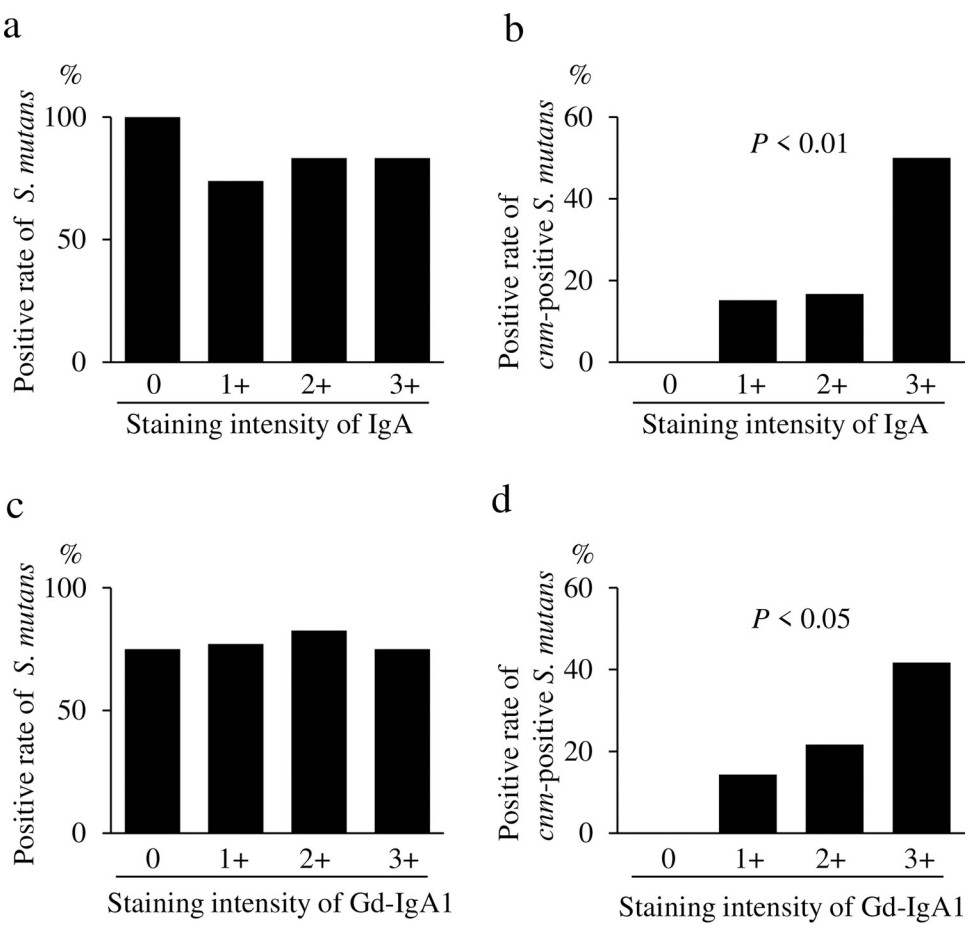

**Fig 2. Association between glomerular staining intensity of IgA and galactose-deficient IgA1 (Gd-IgA1) and the rate of *cnm*-positive *S. mutans* in the oral cavity.** Comparison of the glomerular staining intensity of IgA and the positive rate of *S. mutans* in the oral cavity (a). Comparison of the glomerular staining intensity of IgA and the positive rate of *cnm*-positive *S. mutans* in the oral cavity (b). Comparison of the glomerular staining intensity of Gd-IgA1 (KM55) and the positive rate of *S. mutans* in the oral cavity (c). Comparison of the glomerular staining intensity of Gd-IgA1 (KM55) and the positive rate of *cnm*-positive *S. mutans* in the oral cavity (d). Data were analyzed using the Cochran–Armitage trend test. *P* < 0.05 was considered statistically significant.

associated with Gd-IgA1, these data indicate that *cnm*-positive *S. mutans* in the oral cavity may be one of the factors associated with the development of IgA nephropathy via Gd-IgA1.

Recent clinical studies have demonstrated that *cnm*-positive *S. mutans* in the oral cavity is associated with IgAN [20–22, 26]. One study showed that the positive rate of *cnm*-positive *S. mutans* in the oral cavity was significantly higher in the IgAN group than in the healthy control group (32.1% vs. 14.0%) [20]. However, the positive rate of *S. mutans* was similar between the IgAN and healthy control groups [20]. Another study suggested that *cnm*-positive *S. mutans* in the oral cavity and the dental caries status were associated with urinary protein in patients with IgAN [21]. The Cnm protein in the tonsils may be associated with the severity of IgAN [22]. The report demonstrated the following: (1) the positive Cnm protein area/total tonsillar area rate was significantly higher in patients with IgAN than in the control (chronic tonsillitis) group; (2) Cnm protein existed in the tonsils, not in the glomerulus, in patients with IgAN; and (3) Cnm protein in the tonsils was associated with urinary protein in patients with IgAN [22]. These results suggest that IgAN is aggravated by immune reactions in the tonsils via an unknown mechanism induced by Cnm protein [22]. In rodent models, intravenous

Table 1. Comparison between the *cnm*-positive and *cnm*-negative *S. mutans* groups of patients.

| Characteristics | *cnm*-negative *S. mutans* (n = 59) | *cnm*-positive *S. mutans* (n = 15) | *P* value |
|---|---|---|---|
| **Percentage of staining intensity of IgA: 3+ (%)** | **10.2** | **40** | **0.0051** |
| **Percentage of staining intensity of Gd-IgA1: 3+ (%)** | **11.9** | **33.3** | **0.0440** |
| Age (years) | 44.9 ± 15.3 | 45.1 ± 13.3 | 0.9662 |
| Sex (M/F) | 27/32 | 9/6 | 0.3246 |
| Height (cm) | 162.6 ± 8.6 | 164.3 ± 7.5 | 0.4834 |
| Body weight (kg) | 62.5 ± 13.4 | 64.0 ± 11.3 | 0.6966 |
| BMI (kg/m$^2$) | 23.6 ± 4.5 | 23.7 ± 3.7 | 0.9293 |
| Systolic blood pressure (mmHg) | 125.0 ± 21.3 | 121.5 ± 17.6 | 0.5582 |
| Diastolic blood pressure (mmHg) | 69.2 ± 11.6 | 68.4 ± 8.0 | 0.8046 |
| Serum creatinine (mg/dl) | 1.0 ± 0.7 | 1.0 ± 0.5 | 0.932 |
| eGFR (ml/minute/1.73 m$^2$) | 67.3 ± 26.4 | 67.4 ± 22.6 | 0.9926 |
| Serum C3 (mg/dl) | 108.2 ± 16.1 | 107.0 ± 18.7 | 0.8029 |
| Serum IgA (mg/dl) | 348.3 ± 158.3 | 358.5 ± 164.6 | 0.8255 |
| Urinary protein (g/gCr) | 1.4 ± 0.8 | 0.8 ± 0.7 | 0.1857 |
| Percentage of urinary sediment $\geq$ 100 RBCs/HPF | 8.5 | 26.7 | 0.0543 |

Values are %, number, or mean ± standard deviation. Bold values indicate statistical significance at *P* < 0.05. BMI: body mass index, eGFR: estimated glomerular filtration rate.

administration of *cnm*-positive *S. mutans* induced transient IgAN-like lesions [23] and severe dental caries induced by *cnm*-positive *S. mutans* caused IgAN-like glomerulonephritis [24]. As mentioned above, we found an association between *cnm*-positive *S. mutans* and IgAN. However, we could not identify how *cnm*-positive *S. mutans* induced IgAN.

Gd-IgA1 has been demonstrated as a key effector molecule in the pathogenesis of IgAN [6–9]. We showed an association between the presence of *cnm*-positive *S. mutans* and glomerular Gd-IgA1 in this study. This study indicates that *cnm*-positive *S. mutans* induced Gd-IgA1 production in the process of causing IgAN, suggesting that the most upstream factor for Gd-IgA1 production is oral bacterial infection, such as *cnm*-positive *S. mutans*. We hypothesize that repeated immune reactions involving IgA induced by infection such as *cnm*-positive *S. mutans* in the mucosal immune tissues of the oral cavity (e.g., the tonsils) may cause Gd-IgA1 [21]. We believe that this result is important in considering the pathogenic mechanism of IgAN. We do not have enough evidence to explain how *cnm*-positive *S. mutans* affects the induction of Gd-IgA1 at this stage and need to elucidate these mechanisms in a further study.

We also investigated the positive rate of *C. rectus* and *P. gingivalis* in the oral cavity because these bacteria are reported to be potentially associated with IgA nephropathy [19, 26, 27]. We could not confirm the association between these bacteria and Gd-IgA1 in this research.

Table 2. Analysis of renal biopsy specimens using the Oxford classification.

| Characteristic | *cnm*-negative *S. mutans* (n = 59) | *cnm*-positive *S. mutans* (n = 15) | *P* value |
|---|---|---|---|
| Positive mesangial hypercellularity score (%) | 72.9 | 86.7 | 0.2665 |
| Positive endocapillary hypercellularity score (%) | 50.8 | 46.7 | 0.7725 |
| Positive segmental glomerulosclerosis score (%) | 66.1 | 73.3 | 0.5932 |
| Tubular atrophy/interstitial fibrosis score $\geq$1+ (%) | 43.9 | 46.7 | 0.8565 |
| Crescent score $\geq$1+ (%) | 40.7 | 33.3 | 0.6029 |

Our study showed that the percentage of urinary sediment ≥100 RBCs/high-power field was higher in the cnm-positive *S. mutans* group than in the cnm-negative *S. mutans* group. Although this finding did not reach statistical significance, it suggests that *cnm*-positive *S. mutans* is associated with the hematuria that is the cardinal symptom of IgAN.

Unfortunately, because the exact mechanism of IgAN is not yet understood, there are no specific disease-targeted therapies for IgAN [5]. Our present findings are consistent with the new concept of an oral–kidney association and could establish a new treatment for IgAN [21]. Considering that *S. mutans* is a pathogen that causes dental caries, reducing it in the oral cavity to prevent dental caries may improve the prognosis of IgAN [27]. Further studies are needed to elucidate the effects of oral care as a treatment of IgAN.

There are some limitations in this study. First, although we demonstrated that *cnm*-positive *S. mutans* in the oral cavity was associated with Gd-IgA1 in patients with IgAN, how *cnm*-positive *S. mutans* contributes to Gd-IgA1 needs to be elucidated. Second, given that the only *S. mutans* protein we examined was Cnm, the possibility that other *S. mutans* proteins are associated with Gd-IgA1 cannot be excluded. Third, the sample size was small, and all patients were of the same ethnicity and from a single facility. Further studies in larger numbers of ethnically diverse patients from multiple facilities are needed to confirm our present findings.

In conclusion, *cnm*-positive *S. mutans* in the oral cavity is associated with Gd-IgA1 in glomeruli in patients with IgAN. This finding suggests that *cnm*-positive *S. mutans* is involved in the pathogenicity of IgAN through induction of Gd-IgA1.

## Supporting information

**S1 Fig. Comparison of the glomerular staining intensity of (Gd-IgA1) (KM55) and plasma Gd-IgA1 (KM55) concentrations.** There was no significant association between the glomerular staining intensity of Gd-IgA1 and the plasma Gd-IgA1 concentration. The data were examined for statistical significance using a simple regression analysis. Gd-IgA1: galactose-deficient IgA1.
(TIF)

**S2 Fig. No association between glomerular staining intensity of Gd-IgA1 and the positive rate of *C. rectus* or *P. gingivalis* in the oral cavity.** Comparison of the glomerular staining intensity of IgA and the positive rate of *Campylobacter rectus* in the oral cavity (a). Comparison of the glomerular staining intensity of Gd-IgA1 (KM55) and the positive rate of *C. rectus* in the oral cavity (b). Comparison of the glomerular staining intensity of IgA and the positive rate of *Porphyromonas gingivalis* in the oral cavity (c). Comparison of the glomerular staining intensity of Gd-IgA1 (KM55) and the positive rate of *P. gingivalis* in the oral cavity (d). The data were examined for statistical significance using the Cochran–Armitage trend test.
$P < 0.05$ was considered statistically significant.
(TIF)

## Acknowledgments

## Ethical statement

This study protocol fully adhered to the Declaration of Helsinki (64th WMA General Assembly, Fortaleza, Brazil, 2013). The protocol was approved by the Ethics Committee of Seirei Hamamatsu General Hospital (approval no. 3646), Osaka University Graduate School of Dentistry (approval no. H29-E9), and Okayama University Graduate School of Medicine (approval

no. 1704–036). All patients were informed of the study protocol, and provided written informed consent prior to participating in the study.

## Author Contributions

**Conceptualization:** Hitoshi Suzuki, Kazuhiko Nakano.

**Data curation:** Taro Misaki, Daiki Matsuoka, Seigo Ito, Michiyo Matsumoto-Nakano.

**Formal analysis:** Taro Misaki, Seigo Ito.

**Funding acquisition:** Taro Misaki, Yasuyuki Nagasawa, Kazuhiko Nakano.

**Investigation:** Shuhei Naka, Mingfeng Lee, Ryosuke Aoki, Daiki Matsuoka.

**Methodology:** Taro Misaki, Shuhei Naka, Hitoshi Suzuki, Mingfeng Lee, Ryosuke Aoki, Daiki Matsuoka, Ryota Nomura.

**Project administration:** Shuhei Naka, Hitoshi Suzuki, Michiyo Matsumoto-Nakano, Kazuhiko Nakano.

**Resources:** Taro Misaki, Yasuyuki Nagasawa, Michiyo Matsumoto-Nakano, Kazuhiko Nakano.

**Software:** Shuhei Naka, Yasuyuki Nagasawa, Daiki Matsuoka, Ryota Nomura, Michiyo Matsumoto-Nakano.

**Supervision:** Shuhei Naka, Hitoshi Suzuki, Yasuyuki Nagasawa, Ryota Nomura, Michiyo Matsumoto-Nakano, Yusuke Suzuki, Kazuhiko Nakano.

**Validation:** Hitoshi Suzuki, Yasuyuki Nagasawa, Seigo Ito, Ryota Nomura, Michiyo Matsumoto-Nakano, Yusuke Suzuki, Kazuhiko Nakano.

**Visualization:** Taro Misaki, Yasuyuki Nagasawa, Seigo Ito, Ryota Nomura, Michiyo Matsumoto-Nakano, Yusuke Suzuki, Kazuhiko Nakano.

**Writing – original draft:** Taro Misaki.

**Writing – review & editing:** Kazuhiko Nakano.

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
