## [Decision Letter · Decision Letter 0]

19 Dec 2022

PONE-D-22-21218

cnm -positive Streptococcus mutans is associated with galactose-deficient IgA in patients with IgA nephropathy

PLOS ONE

Dear Dr. Misaki,

Thank you for submitting your manuscript to PLOS ONE. After careful consideration, we feel that it has merit but does not fully meet PLOS ONE’s publication criteria as it currently stands. Therefore, we invite you to submit a revised version of the manuscript that addresses the points raised during the review process.

Specifically, we request you to revise the conclusions, in order to better reflect the results obtained in the study, as indicated by reviewer #1.

In addition, all the comments and corrections indicated by both reviewers need to be considered.

We look forward to receiving your revised manuscript.

Kind regards,

Maria Leonor S Oliveira, PhD

Academic Editor

PLOS ONE

https://journals.plos.org/plosone/s/fileid=ba62/PLOSOne_formatting_sample_title_authors_affiliations.pdf.

3. Thank you for submitting the above manuscript to PLOS ONE. During our internal evaluation of the manuscript, we found significant text overlap between your submission and previous work in the Methods and Discussion sections. We would like to make you aware that copying extracts from previous publications, especially outside the methods section, word-for-word is unacceptable. In addition, the reproduction of text from published reports has implications for the copyright that may apply to the publications. Please revise the manuscript to rephrase the duplicated text, cite your sources, and provide details as to how the current manuscript advances on previous work. Please note that further consideration is dependent on the submission of a manuscript that addresses these concerns about the overlap in text with published work. We will carefully review your manuscript upon resubmission and further consideration of the manuscript is dependent on the text overlap being addressed in full. Please ensure that your revision is thorough as failure to address the concerns to our satisfaction may result in your submission not being considered further.

4. You indicated that you had ethical approval for your study. Please clarify whether minors (patients below the age of 18 years) were included in your study. If yes, in your Methods section, please ensure you have also stated whether you obtained consent from parents or guardians of the minors included in the study or whether the research ethics committee or IRB specifically waived the need for their consent.

“This work was supported by JSPS KAKENHI [grant numbers 19K10098, 21H03149 and 21K08242].”

“This work was supported by JSPS KAKENHI [grant numbers 19K10098 (TM), 21H03149 (KN) and 21K08242 (YN)].

URL of each funder website: https://www.jsps.go.jp/english/index.html

Did the sponsors or funders play any role in the study design, data collection and analysis, decision to publish, or preparation of the manuscript? NO.”

Reviewers' comments:

Reviewer's Responses to Questions

**Comments to the Author**

1. Is the manuscript technically sound, and do the data support the conclusions?

Reviewer #1: Partly

Reviewer #2: Yes

2. Has the statistical analysis been performed appropriately and rigorously? 

Reviewer #1: Yes

Reviewer #2: I Don't Know

3. Have the authors made all data underlying the findings in their manuscript fully available?

Reviewer #1: Yes

Reviewer #2: No

4. Is the manuscript presented in an intelligible fashion and written in standard English?

Reviewer #1: Yes

Reviewer #2: Yes

5. Review Comments to the Author

Reviewer #1: The paper is written in an intelligible way and describes adequately the data obtained by the authors. There are some minor questions related to grammar and clearness of some statements:

line 51: The whole fragment is confuse, the authors should rewrite this line to improve clarity.

line 94: Replace "pathogenesis" for "pathogen"

line 172: The authors should define what TMB solution is.

Although the paper describes clearly the objectives and some findings, the authors were not able to draw a line of evidences which would delineate an horizon to support their hypothesis. Just to be clearer, in lines 230-232, the authors state: "These data indicated that cnm-positive S. mutans was clearly associated with the development of IgA nephropathy via Gd-IgA1."

It is possible that specific Gd-IgA1 directed to Cnm protein would be involved in IgA nephropathy (IgAN) establishment or recrudescence. However, the authors merely showed that IgAN recipients infected with Cnm-positive S. mutans also presented Gd-IgA1 in renal milieu. It represents, undoubtedly, a promising finding which, on the other hand, is not enough to support the hypothesis per se. Therefore, the issues raised in this review are not related to the quality of experiments or the way they were presented, but how the conclusions were drawn in connection with a correlation, which in turn not, necessarily, implicates causality. As a consequence, it seems inadequate to affirm categorically that the presence of Cnm protein is the reason of IgaN status and that the mechanism involves uniquely Gd-IgA1 production in the kidneys, as seen in the fragment mentioned above (lines 230-232).

Co-infection with other bacteria in multispecies biofilms and other physiological indicators, different from those analyzed here, could influence the appearence of IgA. Other factors somewhat ignored cannot be discarded in this context, such as the possibility that other S. mutans proteins could have a role in the pathology. In summary, the authors should have collected other data to support their hypothesis.

In lines 217-219, the authors state: "The percentage of urinary sediment ≥100 RBCs/high-power field was higher in the cnm-positive S. mutans group than in cnm-negative S. mutans group, but this was not significant (Table 1)". The p value of this indicator was very close to the significance level chosen in the analysis. For this reason, the authors should have drawn some inference about this indicator. Above all, statistics indicates tendencies even when the numerical indicator is not strictly between the limits established previously. In this case, this particular indicator should be considered, at least, as a clue to a more complete scenario involving the production of Gd-IgA1, the establishment of IgAN and the presence of S. mutans producing Cnm protein.

Based on these issues, I do not consider this paper suitable for publication in PLoS One. Significant revision, including additional data, are required to justify further consideration of this manuscript in a new submission.

Reviewer #2: The work is interesting and original, the authors have experience in this area. I have several comments regarding the manuscript:

Materials and Methods

In general, I think it is necessary to check if all information is described.

1. Histological studies (line131-132): What are the fluorochromes used?

2. Statistical analysis: I'm not an expert in statistics but it seems to me that it's not the best to do statistical analysis from results that are in percentage (Fig 2).

Results

1. Fig 1. I would like to see images of glomerular staining intensity slices at a lower magnification. The objective would be to have a better view of how frequent this glomerular staining intensity is in renal biopsy samples.

2. Fig 2. The data shown are in percentage and as mentioned earlier (M&M) are significant between groups.

3. Figure Legends: Legends are incomplete. Fig1 is missing data such as number of samples per group, antibodies and fluorochromes used. Fig2 is missing the significance value.

4. Tables: Table 1. Data with significance from data represented in percentage. Table 2. It does not say if the data are in percentage. You could put the number of patients by characteristics analyzed.

Discussion

1. In the second paragraph, change the sentence of line 233.

Supplementary Figure

The supplementary figure must be submitted the same quality as the other figures in the paper. The legend is incomplete.

6. PLOS authors have the option to publish the peer review history of their article (what does this mean?). If published, this will include your full peer review and any attached files.

Reviewer #1: No

Reviewer #2: No

---

## [Author Response · Author response to Decision Letter 0]

10 Jan 2023

1. Please ensure that your manuscript meets PLOS ONE's style requirements, including those for file naming. The PLOS ONE style templates can be found at …

Response: We thank the editorial staff of PLOS ONE for pointing this out. We have formatted the manuscript according to the instructions provided by PLOS ONE.

Response: We appreciate this advice. We have now included our tables in the main manuscript and removed the individual files.

3. Thank you for submitting the above manuscript to PLOS ONE. During our internal evaluation of the manuscript, we found significant text overlap between your submission and previous work in the Methods and Discussion sections. We would like to make you aware that copying extracts from previous publications, especially outside the methods section, word-for-word is unacceptable. In addition, the reproduction of text from published reports has implications for the copyright that may apply to the publications. Please revise the manuscript to rephrase the duplicated text, cite your sources, and provide details as to how the current manuscript advances on previous work. Please note that further consideration is dependent on the submission of a manuscript that addresses these concerns about the overlap in text with published work. We will carefully review your manuscript upon resubmission and further consideration of the manuscript is dependent on the text overlap being addressed in full. Please ensure that your revision is thorough as failure to address the concerns to our satisfaction may result in your submission not being considered further.

Response: We are grateful for these comments. We have revised our manuscript so that it does not overlap with any text from previous publications. The sections in green font are revised text. In the Methods section, the parts of our experimental protocol that were similar to those in previous experiments have been acknowledged and referenced. 

4. You indicated that you had ethical approval for your study. Please clarify whether minors (patients below the age of 18 years) were included in your study. If yes, in your Methods section, please ensure you have also stated whether you obtained consent from parents or guardians of the minors included in the study or whether the research ethics committee or IRB specifically waived the need for their consent.

Response: We confirm that our study did not include patients younger than 18 years. We have added the following explanatory text in the Methods section:

“All patients were over 18 years of age.” (lines 102–103)

“This work was supported by JSPS KAKENHI [grant numbers 19K10098, 21H03149 and 21K08242].”

“This work was supported by JSPS KAKENHI [grant numbers 19K10098 (TM), 21H03149 (KN) and 21K08242 (YN)].

URL of each funder website: https://www.jsps.go.jp/english/index.html

Did the sponsors or funders play any role in the study design, data collection and analysis, decision to publish, or preparation of the manuscript? NO.”

Response: We apologize for this error and have deleted the funding information in the Acknowledgment section. The Funding statement should read as follows: “This work was supported by JSPS KAKENHI (grant numbers 19K10098 [TM], 21H03149 [KN] and 21K08242 [YN]). The sponsors or funders did not play any role in the study design, data collection and analysis, decision to publish, or preparation of the manuscript.”

Response: We are grateful for these comments. Although we could open up the data, we wish to change our Data Availability statement to “No”.

Reviewers' comments:

Reviewer's Responses to Questions

Comments to the Author

3. Have the authors made all data underlying the findings in their manuscript fully available?

Reviewer #1: Yes

Reviewer #2: No

Response: We understand the response of Reviewer 2 to be related to his/her question: “Statistical analysis: I'm not an expert in statistics but it seems to me that it's not the best to do statistical analysis from results that are in percentage (Fig 2).”

We did consider various statistical methods for analysis of our study data. However, the results for the bacteria evaluated in this study were either positive or negative and there was no other method that could be used to evaluate the positivity rate. Therefore, we believe it is appropriate to express the positivity rate as a percentage.

5. Review Comments to the Author

Reviewer #1: The paper is written in an intelligible way and describes adequately the data obtained by the authors. There are some minor questions related to grammar and clearness of some statements:

line 51: The whole fragment is confuse, the authors should rewrite this line to improve clarity.

Response: We apologize for the lack of clarity of the abovementioned text. We have revised it to read as follows:

”The presence of S. mutans and cnm-positive S. mutans was evaluated in saliva samples of 74 patients with IgAN or IgA vasculitis by polymerase chain reaction.” (lines 43–45)

line 94: Replace "pathogenesis" for "pathogen"

Response: We apologize for this typographical error and have corrected it (line 87).

line 172: The authors should define what TMB solution is.

Response: We have expanded “TMB” to “tetramethylbenzidine” (line 160).

Although the paper describes clearly the objectives and some findings, the authors were not able to draw a line of evidences which would delineate an horizon to support their hypothesis. Just to be clearer, in lines 230-232, the authors state: "These data indicated that cnm-positive S. mutans was clearly associated with the development of IgA nephropathy via Gd-IgA1."

It is possible that specific Gd-IgA1 directed to Cnm protein would be involved in IgA nephropathy (IgAN) establishment or recrudescence. However, the authors merely showed that IgAN recipients infected with Cnm-positive S. mutans also presented Gd-IgA1 in renal milieu. It represents, undoubtedly, a promising finding which, on the other hand, is not enough to support the hypothesis per se. Therefore, the issues raised in this review are not related to the quality of experiments or the way they were presented, but how the conclusions were drawn in connection with a correlation, which in turn not, necessarily, implicates causality. As a consequence, it seems inadequate to affirm categorically that the presence of Cnm protein is the reason of IgAN status and that the mechanism involves uniquely Gd-IgA1 production in the kidneys, as seen in the fragment mentioned above (lines 230-232).

Co-infection with other bacteria in multispecies biofilms and other physiological indicators, different from those analyzed here, could influence the appearance of IgA. Other factors somewhat ignored cannot be discarded in this context, such as the possibility that other S. mutans proteins could have a role in the pathology. In summary, the authors should have collected other data to support their hypothesis.

Response: We thank the reviewer for these valuable observations and agree with them.

In this study, we demonstrated an association between the presence of cnm-positive S. mutans in the oral cavity and glomerular Gd-IgA1. However, we acknowledged that other factors could be involved in the pathogenesis of Gd-IgA1.

We have performed an additional experiment to determine the P. gingivalis and C. rectus positivity rates in the oral cavity in view of the findings of previous studies performed by our group that suggest these bacteria may be associated with IgA nephropathy (Nagasawa Y et al., Int J Mol Sci. 2021;22(23), Misaki T et al., Nephron, 2018;139(2):143-9). We have added the findings of this additional experiment as S2 Fig a-d. We found no association between these two bacteria and Gd-IgA1. We have added text in the Methods section (lines 117–118), Results section (lines 195–200), Discussion section (lines 258–261) and the Fig legends in the Supportive information S2 that provides the details of this additional experiment.

We have also included the following explanatory text in the Discussion section: 

“Although various factors are considered to be associated with Gd-IgA1, these data indicate that cnm-positive S. mutans in the oral cavity may be one of the factors associated with the development of IgA nephropathy via Gd-IgA1.” (lines 224–227)

We have also added the following text in the paragraph addressing the limitations of the study in the Discussion section:

“Second, given that the only S. mutans protein we examined was Cnm, the possibility that other S. mutans proteins are associated with Gd-IgA1 cannot be excluded.” (lines 275–277).

In lines 217-219, the authors state: "The percentage of urinary sediment ≥100 RBCs/high-power field was higher in the cnm-positive S. mutans group than in cnm-negative S. mutans group, but this was not significant (Table 1)". The p value of this indicator was very close to the significance level chosen in the analysis. For this reason, the authors should have drawn some inference about this indicator. Above all, statistics indicates tendencies even when the numerical indicator is not strictly between the limits established previously. In this case, this particular indicator should be considered, at least, as a clue to a more complete scenario involving the production of Gd-IgA1, the establishment of IgAN and the presence of S. mutans producing Cnm protein.

Based on these issues, I do not consider this paper suitable for publication in PLoS One. Significant revision, including additional data, are required to justify further consideration of this manuscript in a new submission.

Response: We are very grateful to the reviewer for these helpful comments. We were hesitant to discuss non-significant events, so did not do so. However, given our view that cnm-positive S. mutans exacerbates IgAN, we consider that the trend of increased hematuria in the cnm-positive S. mutans group is an important finding. Accordingly, we have added the following text in the Discussion section:

“Our study showed that the percentage of urinary sediment ≥100 RBCs/high-power field was higher in the cnm-positive S. mutans group than in the cnm-negative S. mutans group. Although this finding did not reach statistical significance, it suggests that cnm-positive S. mutans is associated with the hematuria which is the one symptom of the IgAN.” (lines 262–266)

Reviewer #2: The work is interesting and original, the authors have experience in this area. I have several comments regarding the manuscript:

Materials and Methods

In general, I think it is necessary to check if all information is described.

1. Histological studies (line131-132): What are the fluorochromes used?

Response: We appreciate the reviewer’s prompt and have added the name of the IF antibody used as follows:

“ For the immunofluorescence analysis, frozen sections were subjected to fluorescence by fluorescein-conjugated goat IgG fraction to human IgG (F110FC, American Qualex, San Clemente, CA, USA), fluorescein-conjugated goat IgG fraction to human IgA (55077, MP Biomedicals, Solon, OH, USA), fluorescein-conjugated goat IgG fraction to human IgM (55153, MP Biomedicals), fluorescein-conjugated goat IgG fraction to human C3 (55167, MP Biomedicals), fluorescein-conjugated rabbit anti human C1q (F0254, DAKO Japan Inc., Kyoto, Japan), and fluorescein-conjugated goat IgG fraction to human fibrinogen (55169, MP Biomedicals). (lines 124–132).

2. Statistical analysis: I'm not an expert in statistics but it seems to me that it's not the best to do statistical analysis from results that are in percentage (Fig 2).

Response: We appreciate this comment regarding our statistical analysis. We considered alternative statistical methods for analysis of our study data. However, the results for the bacteria evaluated in this study were either positive or negative, and there was no other method that could be used to evaluate the positivity rate. Therefore, we consider it appropriate to express the positivity rate as a percentage.

Results

1. Fig 1. I would like to see images of glomerular staining intensity slices at a lower magnification. The objective would be to have a better view of how frequent this glomerular staining intensity is in renal biopsy samples.

Response: We thank the reviewer for this comment and agree with it. We examined multiple glomeruli in each case and confirmed that several glomeruli had similar staining intensity in each case. In clinical practice, it is generally accepted that the diagnosis is made by IF staining of a single glomerulus. We have presented an example of confirmation in multiple glomeruli for the reviewer only (Figure 20221222). Each glomerulus had similar staining intensity.

2. Fig 2. The data shown are in percentage and as mentioned earlier (M&M) are significant between groups.

Response: We are grateful to the reviewer for this observation. As mentioned in our response to Question 2 posed by Reviewer 2, we considered various statistical methods for analysis of our study data. However, the results for the bacteria evaluated in our study were either positive or negative, and there was no alternative method that could be used to evaluate the positivity rate. Therefore, we consider it appropriate to express the positivity rate as a percentage.

3. Figure Legends: Legends are incomplete. Fig1 is missing data such as number of samples per group, antibodies and fluorochromes used. Fig2 is missing the significance value.

Response: Seventy-four patients were divided into four groups according to the staining intensity of IgA: group 0 (n = 4), group 1+ (n = 46), group 2+ (n = 12), and group 3+ (n = 12). These patients were also divided into four groups according to the staining intensity of Gd-IgA1: group 0 (n = 4), group 1+ (n = 35), group 2+ (n = 23), and group 3+ (n = 12). Fluorescein–conjugated polyclonal rabbit anti-human IgA antibody (DAKO Japan Inc., Kyoto, Japan), anti-human Gd-IgA1 antibody (KM55) (Immuno-Biological Laboratories Co., Ltd, Gunma, Japan), and Alexa Fluor 555-conjugated goat anti-rat IgG antibody (Life Technologies) was used for immunofluorescence staining.

We have also added the following text in the legend to Figure 2:

“P < 0.05 was considered statistically significant.”

4. Tables: Table 1. Data with significance from data represented in percentage. Table 2. It does not say if the data are in percentage. You could put the number of patients by characteristics analyzed.

Response: We agree with the reviewer’s suggestion regarding consistency of reporting. We have revised Table 2 so that the data therein are now expressed as percentages, as in Table 1. In view of the difference in the number of patients in the two groups, we think it appropriate to express the data as percentages only. The number of patients in each group can be calculated easily by checking the number of patients in the column headed “Characteristics”.

Discussion

1. In the second paragraph, change the sentence of line 233.

Response: We thank the reviewer for pointing out the awkward text in line 233. We have changed it from “Recently, the results of recent clinical studies have suggested that cnm-positive S. mutans in the oral cavity is associated with IgAN.” to “Recent clinical studies have suggested that cnm-positive S. mutans in the oral cavity is associated with IgAN.” (lines 228–229).

Supplementary Figure

The supplementary figure must be submitted the same quality as the other figures in the paper. The legend is incomplete.

(Response)

We have revised the abovementioned figure, completed its legend, and included it as Supporting Information in the text.

---

## [Decision Letter · Decision Letter 1]

14 Feb 2023

cnm -positive Streptococcus mutans is associated with galactose-deficient IgA in patients with IgA nephropathy

PONE-D-22-21218R1

Dear Dr. Misaki,

We’re pleased to inform you that your manuscript has been judged scientifically suitable for publication and will be formally accepted for publication once it meets all outstanding technical requirements.

Kind regards,

Maria Leonor S Oliveira, PhD

Academic Editor

PLOS ONE

Additional Editor Comments (optional):

Reviewers' comments:

Reviewer's Responses to Questions

**Comments to the Author**

1. If the authors have adequately addressed your comments raised in a previous round of review and you feel that this manuscript is now acceptable for publication, you may indicate that here to bypass the “Comments to the Author” section, enter your conflict of interest statement in the “Confidential to Editor” section, and submit your "Accept" recommendation.

Reviewer #2: All comments have been addressed

2. Is the manuscript technically sound, and do the data support the conclusions?

Reviewer #2: Yes

3. Has the statistical analysis been performed appropriately and rigorously? 

Reviewer #2: I Don't Know

4. Have the authors made all data underlying the findings in their manuscript fully available?

Reviewer #2: Yes

5. Is the manuscript presented in an intelligible fashion and written in standard English?

Reviewer #2: Yes

6. Review Comments to the Author

Reviewer #2: (No Response)

7. PLOS authors have the option to publish the peer review history of their article (what does this mean?). If published, this will include your full peer review and any attached files.

Reviewer #2: No

---

## [Editor Report · Acceptance letter]

17 Feb 2023

PONE-D-22-21218R1 

*cnm*-positive *Streptococcus mutans* is associated with galactose-deficient IgA in patients with IgA nephropathy 

Dear Dr. Misaki:

I'm pleased to inform you that your manuscript has been deemed suitable for publication in PLOS ONE. Congratulations! Your manuscript is now with our production department. 

Kind regards, 

on behalf of

Dr. Maria Leonor S Oliveira 

Academic Editor

PLOS ONE